# *Rosmarinus officinalis* L. Leaf Extracts and Their Metabolites Inhibit the Aryl Hydrocarbon Receptor (AhR) Activation In Vitro and in Human Keratinocytes: Potential Impact on Inflammatory Skin Diseases and Skin Cancer

**DOI:** 10.3390/molecules27082499

**Published:** 2022-04-13

**Authors:** Panagiotis Kallimanis, Ioanna Chinou, Angeliki Panagiotopoulou, Anatoly A. Soshilov, Guochun He, Michael S. Denison, Prokopios Magiatis

**Affiliations:** 1Laboratory of Pharmacognosy and Chemistry of Natural Products, Department of Pharmacy, National and Kapodistrian University of Athens, Panepistimiopolis Zografou, 15771 Athens, Greece; pgkallimanis@gmail.com; 2Institute of Biosciences and Applications, National Centre for Scientific Research “Demokritos”, Patr. Gregoriou E & 27 Neapoleos Str., Agia Paraskevi, 15310 Athens, Greece; apanagio@bio.demokritos.gr; 3Department of Environmental Toxicology, University of California, Davis, CA 95616, USA; anatoly.soshilov@oehha.ca.gov (A.A.S.); gchhe@ucdavis.edu (G.H.); msdenison@ucdavis.edu (M.S.D.)

**Keywords:** *Rosmarinus officinalis* L., AhR, TCDD, FICZ, pityriazepin, indirubin, seborrheic dermatitis, human keratinocytes, skin cancer

## Abstract

Aryl hydrocarbon receptor (AhR) activation by environmental agents and microbial metabolites is potentially implicated in a series of skin diseases. Hence, it would be very important to identify natural compounds that could inhibit the AhR activation by ligands of microbial origin as 6-formylindolo[3,2-b]carbazole (FICZ), indirubin (IND) and pityriazepin (PZ) or the prototype ligand 2,3,7,8-tetrachlorodibenzo-*p*-dioxin (TCDD). Five different dry *Rosmarinus officinalis* L. extracts (ROEs) were assayed for their activities as antagonists of AhR ligand binding with guinea pig cytosol in the presence of [^3^H]TCDD. The methanolic ROE was further assayed towards CYP1A1 mRNA induction using RT-PCR in human keratinocytes against TCDD, FICZ, PZ, and IND. The isolated metabolites, carnosic acid, carnosol, 7-*O*-methyl-*epi*-rosmanol, 4′,7-*O*-dimethylapigenin, and betulinic acid, were assayed for their agonist and antagonist activity in the presence and absence of TCDD using the gel retardation assay (GRA). All assayed ROE extracts showed similar dose-dependent activities with almost complete inhibition of AhR activation by TCDD at 100 ppm. The methanol ROE at 10 ppm showed 99%, 50%, 90%, and 85% inhibition against TCDD, FICZ, IND, and PZ, respectively, in human keratinocytes. Most assayed metabolites exhibited dose-dependent antagonist activity. ROEs inhibit AhR activation by TCDD and by the *Malassezia* metabolites FICZ, PZ, and IND. Hence, ROE could be useful for the prevention or treatment of skin diseases mediated by activation of AhR.

## 1. Introduction

The aryl hydrocarbon receptor (AhR, also called dioxin receptor) is a ligand-dependent transcription factor with multifaceted biological functions. AhR responds to exogenous and endogenous chemicals by inducing/repressing the expression of several genes with toxic or protective effects in a wide range of species and tissues. The fact that AhR is expressed in all skin cell types suggests a role for AhR signaling at this barrier organ [1,2,3,4]. Indeed, AhR participates in the maintenance of skin homeostasis [5], the enhancement of wound healing, and the partial mediation of ultraviolet radiation damage through the photochemical intracellular production of the potent AhR ligand, 6-formylindolo[3,2-b]carbazole (FICZ) [6].

Furthermore, many studies have shown that AhR seems to be involved in the pathogenesis of some skin diseases, even if the currently available data are contradictory: there are studies that showed a correlation between AhR activation and skin inflammatory diseases [7,8,9,10,11], while, in others, the AhR activation seems to be advantageous for the treatment of inflammatory skin diseases [3,4,12].

The situation seems to be less complicated in the case of skin cancer. Indeed, blocking the AhR signaling activity could prevent or treat skin cancer [6,11,13,14,15,16,17,18,19,20,21,22]. It is well known that the 2,3,7,8-tetrachlorodibenzo-*p*-dioxin (TCDD) causes many of its detrimental effects through persistent activation of this receptor and the downstream signaling pathway [23]. Studies using knockout mice demonstrated that TCDD toxicity is almost solely dependent upon a functional AhR [24,25]. FICZ, derived from tryptophan metabolism after exposure to UV radiation of the sun, seems to be involved in skin cancer via the AhR activation [16]. FICZ and indirubin (IND) produced by *Malassezia* may also be implicated in skin carcinogenesis [11,18].

The appearance of chloracne, a characteristic acne-like eruption, is one of the heralding signs of TCDD intoxication in humans [8]. The strong and mainly the prolonged activation of AhR by dioxins results in excessive hyperkeratinization of keratinocytes and sebocytes, leading to chloracne [26,27].

Activation of the AhR plays an important role in mediating the biological/toxicological effects of a variety of environmental xenobiotics, including the *Malassezia* produced IND and indolo[3,2-b] carbazole (ICZ) on the immune system [28,29]. In seborrheic dermatitis (SD), a chronic inflammatory dermatosis in which pathogenesis *Malassezia* spp. seems to play a crucial role, the activators of AhR, FICZ, ICZ, IND, and pityriazepin (PZ) have been found to be produced by *Malassezia furfur* in higher amounts in scales of a lesioned area than in healthy skin [9,11,30].

Therefore, for its multifaceted role in skin homeostasis and diseases, AhR seems to be an attractive therapeutic target, and it would be of importance to identify natural compounds that could mediate their beneficial biological effects via AhR for the treatment of inflammatory skin diseases and potentially the prevention or treatment of skin cancer. For example, compounds that inhibit the AhR activation by environmental ligands such as pollutants, such as TCDD, and of microbial origins, such as FICZ and IND, could be useful to prevent or treat skin diseases.

Rosemary (*Rosmarinus officinalis* L., RO) is a shrub of the Lamiaceae family, which has been widely used in the Mediterranean area since antiquity in folk medicine and as a culinary spice. Almost all over the world, RO leaves are used as decoctions, infusions, and tinctures to cure or prevent a wide range of health disorders, while currently, it has been widely investigated as a food additive [31].

Topical application in mouse skin of *R. officinalis* L. extract (ROE) and its constituent carnosol (CS) inhibited tumor initiation and promotion [32]. ROE reduced the proliferation of the human melanoma A375 cell line [33], while CS inhibited mutagenesis by benzo[a]pyrene in cell lines derived from the skin (HaCaT) [34].

Previously obtained data from patients have shown that RO can express a therapeutic effect on inflammatory skin diseases such as seborrheic dermatitis. Indeed, the topical application of an adequate pharmaceutical form containing *Rosmarinus officinalis* L. leaf extract in male and female subjects suffering from facial and/or scalp seborrheic dermatitis has been found to offer the clinical clearance of this dermatosis in a short time and without side effects [35].

All the above led us to investigate the potential correlation between ROE and its main components with AhR thoroughly and to further explore whether ROE protects from the toxic effects of TCDD, the prototypical environmental pollutant, the most common dioxin and a potent tissue damaging agent [36], and also FICZ, IND and PZ, the most powerful AhR ligands of microbial origin.

For this purpose: (1) Five different ROEs were assayed for their activities as antagonists of AhR ligand binding with guinea pig cytosol in the presence of [^3^H]TCDD, (2) a specific methanolic ROE was further assayed towards CYP1A1 mRNA induction using RT-PCR, in human keratinocytes against TCDD, FICZ, PZ and IND (3) the major metabolites of the specific methanolic ROE, namely, carnosic acid (CA), carnosol (CS), 7-*O*-methyl-epi-rosmanol(7MER), 4′,7-*O*-dimethylapigenin (DMA), and betulinic acid (BA), were assayed for their potential agonist and/or antagonist activity in the presence and absence of TCDD with the method of gel retardation assay (GRA).

## 2. Results

### 2.1. Rosemary Extracts Inhibit TCDD-Induction of AhR-Dependent Reporter Gene Expression

To examine the ability of rosemary extracts to interact with the AhR signaling pathway, we first determined their ability to induce AhR-dependent gene expression. Recombinant human hepatoma (HG2L7.5c1) cells, which contain an AhR-responsive luciferase reporter plasmid under the control of 20 dioxin responsive elements (DREs), respond to AhR agonists with the induction of firefly luciferase in a time-, dose- and AhR-dependent manner [37]. While TCDD induced luciferase activity in HG2L7.5c1 cells, no induction was observed in cells incubated with 100 ppm of the rosemary extracts (R1–R5) (Figure 1).

In fact, a reduction in luciferase activity below that of the DMSO control was observed. Given the lack of any visually observed toxicity in rosemary extract-treated cells and the low levels of AhR-dependent luciferase activity in the absence of added exogenous AhR agonists (i.e., due to endogenous ligands and/or other mechanisms), the extracts could be acting as AhR antagonists and/or that they repress gene expression by a different mechanism (i.e., an inhibitory effect on transcription and or protein synthesis). In order to determine whether the rosemary extracts could antagonize AhR-dependent gene expression, HG2L7.5c1 cells were incubated with TCDD in the absence and presence of 100 ppm of each extract (Figure 1), and these analyses revealed that each of the rosemary extracts could inhibit TCDD-induction of AhR-dependent reporter gene activity. However, these results do not differentiate between whether the rosemary extracts are acting as AhR antagonists or repressing induction of luciferase activity by an AhR-independent mechanism.

### 2.2. Rosemary Extracts Inhibit TCDD-Stimulated AhR Transformation and DNA Binding

One avenue to test whether the rosemary extracts are AhR antagonists is to demonstrate that these extracts can inhibit TCDD-dependent transformation and DNA binding of the AhR in vitro. Accordingly, guinea pig hepatic cytosolic AhR was incubated with TCDD in the absence or presence of two different concentrations of each rosemary extract (10 and 100 ppm), and their effect on TCDD-inducible AhR-DNA binding was examined by gel retardation analysis (Figure 2). These results revealed that all of the rosemary extracts could significantly decrease the amount of TCDD-inducible AhR:DNA binding complex in a concentration-dependent manner, with 10 and 100 ppm of rosemary extracts reducing the amount of induced protein-DNA complex by between 22–33% and 85–92%, respectively. Together, the above results are consistent with the ability of all five rosemary extracts to antagonize/inhibit the ligand-dependent transformation and DNA binding of the AhR, leading to the inhibition of TCDD-inducible, AhR-dependent reporter gene expression shown in Figure 1.

### 2.3. Rosemary Extracts Compete with [^3^H]TCDD for Specific Binding to the AhR

Antagonism of AhR signaling by the ROEs is most likely due to the ability of a chemical(s) present in each extract to act as a classical antagonist and bind to its ligand-binding site in a nonproductive manner thus inhibiting AhR functionality. However, chemicals in these extracts could antagonize AhR signaling by an allosteric mechanism(s) that is mediated via a distinctly different interaction with the AhR or with AhR-associated proteins that negatively affect AhR transformation and/or DNA binding (e.g., adversely affecting ligand-dependent dissociation of AhR-bound proteins (i.e., hsp90, XAP2, p23) and/or AhR dimerization with ARNT or another mechanism). In order to determine whether chemicals in the rosemary extracts are direct antagonists that bind to the AhR ligand binding site, their ability to competitively inhibit the specific binding of [^3^H]TCDD to cytosolic AhR was examined using the hydroxyapatite binding assay. The results in Figure 3 reveal that all five ROEs contained a chemical(s) that could directly compete with [^3^H]TCDD for specific binding to the AhR, and this competition occurred in a concentration-dependent manner (with 10 and 100 ppm of rosemary extracts reducing the amount of [^3^H]TCDD specific binding by between 28–40% and 85–100%, respectively). Thus, all five ROEs contain a chemical(s) that directly bind to and antagonize the AhR and AhR signaling.

### 2.4. Rosemary Extract R4 Antagonizes Induction of CYP1A1 by TCDD and Other AhR Agonists

In order to begin the identification of the AhR antagonists present in ROEs, all extracts were analyzed by quantitative ^1^H-NMR spectra in CDCl_3_. The R4 extract was selected for further biochemical and chemical analyses as it presented the richest metabolic profile and the highest content regarding carnosol and also the strongest activity among all studied samples. Before further fractionation of R4, not only to confirm that this extract could also antagonize the induction of an endogenous AhR-responsive gene (CYP1A1), but that the antagonism was not selective for TCDD and that it would also antagonize other AhR agonists. This latter issue is significant since previous studies have led to the identification of an AhR antagonist (CH223191) that could selectively antagonize TCDD and related halogenated aromatic hydrocarbons, but not that of polycyclic aromatic hydrocarbon (PAH) and PAH-like AhR agonists, such as beta-naphthoflavone [38]. The ability of ROE R4 to antagonize TCDD-inducible CYP1A1 gene induction in human keratinocyte (SIK 28) cells was confirmed by demonstrating an increase in mRNA levels using RT-PCR. Incubation of SIK 28 cells with TCDD for 2 h resulted in a significant increase in CYP1A1 mRNA levels above that of DMSO, and this induction was inhibited by co-incubation of cells with TCDD and extract R4 (Figure 4A); R4 alone had no apparent effect on basal CYP1A1 mRNA levels. ROE-4 was also able to antagonize induction of CYP1A1 mRNA levels by the AhR agonists PZ, FICZ, and IND (Figure 4B), demonstrating that the antagonist(s) in the rosemary R4 extract did not appear to be ligand selective. Thus, the R4 extraction method was used for further isolation and characterization of AhR active secondary metabolites in rosemary.

### 2.5. Identification and Characterization of AhR Active Metabolites from ROEs

When utilizing the same method of extraction and methanol as a solvent for various times of maceration, the following compounds were isolated and quantified from the dry leaves of RO using chromatographic and NMR methods: 4′,7-*O*-dimethylapigenin (**1**), 7-*O*-methyl-*epi*-rosmanol (**2**), carnosol (**3**), carnosic acid (**4**) and betulinic acid (**5**) (Figure 5).

The quantification results are presented in Appendix A. In order to evaluate the ability of these compounds to stimulate or inhibit AhR transformation and DNA binding, the AhR activity was measured using gel retardation analysis. Guinea pig hepatic cytosol was incubated with DMSO, TCDD (at a concentration of 20 nM), or each of the isolated chemicals (at a concentration of 10 μM and 100 μM) to evaluate agonist activity, or a combination of TCDD and the indicated chemical (at both concentrations) to evaluate their antagonist activity. An aliquot of the total R4 extract was included as a reference control. These analyses revealed (Figure 6A,B) that each of the test compounds could antagonize TCDD-stimulated AhR transformation and DNA binding in a concentration-depending manner, similar to that of extract R4. At 10 μM, the inhibition by **one**, **two**, **three**, **four**, and **five** were between 20–63% and between 67–83% at 100 μM. Interestingly, although **one** and **two** also exhibited low levels of AhR agonist activity, stimulating AhR transformation and DNA binding to between 15–18% for **two** and 31–38% for **one**, there was no concentration-dependent increase in the agonist induction response. The lack of a concentration-dependent increase by these compounds is not surprising and likely results from the balance of both agonist and antagonist activities of these compounds. Thus, these results demonstrate that **three**, **four**, and **five** are AhR antagonists, while **one** and **two** are partial agonists, which exhibit partial antagonist activity.

## 3. Discussion

### 3.1. ROE and AhR

Aryl hydrocarbon receptor (AhR), normally found in its inactive form in the cytoplasm, can be activated through two distinct pathways, the canonical and the non-canonical, respectively. The inactive form is maintained through bonding with chaperone proteins (p23, HSP90, AIP/ARA9, or XAP-2) [39]. The canonical pathway starts when an agonist binds to AhR after diffusing through the plasma membrane. The chaperone proteins are released, and the AhR-agonist compound enters the nucleus, where it will bind to the xenobiotic-response elements (XPEs) [40]. Those elements are found in the promoter regions of various genes of the DNA and thus is modulated the expression of downstream genes (phase I, e.g., CYP1A1, and phase II, e.g., UGT1A1) and the AhR repressor (AhRR), which downregulates AhR signaling [39,40,41].

In the non-canonical pathway, AhR is activated by diverse compounds, and it interacts with other receptor-mediated signaling pathways. Structural diversity of the activating ligands along with the cell type, the tissue, and/or some environmental factors may involve AhR activation in cell cycle regulation, mitogen-activated protein kinase cascades, cross-talk with other nuclear receptors, and immediate-early gene induction [42,43,44].

Ligands that can bind to and activate AhR include specific hydrophobic environmental contaminants, such as halogenated and non-halogenated polycyclic aromatic hydrocarbons, such as dioxins and benzo[a]pyrene [1,2]. Various phytochemicals, such as quercetin, resveratrol, and curcumin, are proved to be indirect activators of AhR [45]. Moreover, *Malassezia* metabolites [9,11,30,46] and photo-induced chemicals [47,48] are among the best-characterized high-affinity ligands.

Patients with seborrheic dermatitis (SD), a *Malesezzia* associated skin disease, were found to have in their skin extracts a 10–1000-fold higher capacity for AhR activation compared to control. In these patients’ skin extracts, the presence of IND, FICZ, ICZ, malassezin, and pityriacitrin was demonstrated [11]. It is believed that *Malasezzia* yeasts could participate in the in loco synthesis of these highly potent AhR activators, disturbing skin homeostasis and causing or contributing to the appearance of the disease [49]. Exacerbations of SD, as well as atopic dermatitis (AD), an inflammatory skin condition, are also associated with *Malasezzia* yeast [49]. Several indolic substances, synthesized in vitro by *M. furfur*, have been identified as possible pathogenic AhR activators related to skin disease [50].

Constitutive AhR activation has been shown to be related to inflammatory skin disease after exposure to occupational or environmental xenobiotics [7]. This conclusion is consistent with epidemiologic data showing that environmental pollution with specific AhR activators (polycyclic aromatic hydrocarbons, PAHS) is related to higher numbers of eczema appearance in patients [10]. The implication of AhR in AD is supported by both mouse model studies [51] as well as shown by upregulated AhR in patients with AD compared to healthy people [52]. Hence, in the case of SD, the activation of AhR from the indoles produced by *M. furfur* has been proposed to have a critical role in disease pathogenesis [9,11].

Natural extracts and certain secondary metabolites are particularly interesting because they usually have low toxicity and combine multiple synergistic effects. *R. officinalis* L. was considered a potential source of extracts and ingredients that could have an effect on the inhibition of AhR activation and consequently a protective role in a series of skin diseases. In fact, this hypothesis was based on previously obtained data from patients with inflammatory skin diseases such as seborrheic dermatitis. *R. officinalis* L. extracts have been found to offer clinical clearance of this dermatosis in a short time and without side effects [35].

All these data led us to investigate the behavior of *R. officinalis* L. leaf extract and its main isolated metabolites, 4′,7-*O*-dimethylapigenin (**1**), 7-*O*-methyl-epi-rosmanol (**2**), carnosol (**3**), carnosic acid (**4**) and betulinic acid (**5**) on AhR. This study shows that a specifically prepared ROE acts as an AhR antagonist with respect to TCDD, PZ, IND, and FICZ in human skin cells.

The specifically prepared RO leaf extract and its constituents CA, CS, and BA as AhR antagonists regarding TCDD could be further potentially used for the prevention and/or treatment of the dioxin’s toxicity in various systems such as: immune, reproductive, endocrine, nervous and in the case of chloracne as well.

A RO extract rich in CA and CS increased Nrf2 levels in vitro in HepG2 cells and in vivo in cavies [53,54]. Moreover, CA, CS, and BA along are able to activate Nrf2 [54,55,56,57,58,59,60]. Taking together these results with ours, we conclude that *R. officinalis* L. extracts as well as CA, CS, and BA belong to the AhR antagonist/Nrf2 agonist category. These compounds are potentially strong candidates for preventing the serious health effects caused by toxic environmental agents such as dioxins and, in general, in oxidation conditions. Inhibition of AhR may reduce the activation of CYP1-mediated chemicals, while the antioxidant defense initiated by Nrf2 stimulation may simultaneously remove another reactive metabolite [26,61].

To our knowledge, here it is presented for the first time that a specifically prepared ROE is an antagonist of AhR that can attenuate the actions of TCDD, FICZ, PZ, and IND on this receptor and potentially could be a new pharmacological approach to diseases involving the mentioned AhR-activating indole derivates of *M. furfur* and the pollutant TCDD. Moreover, the isolated metabolites: flavone 4′,7-*O*-dimethylapigenin (**1**), the abietane derivatives 7-*O*-methyl-*epi*-rosmanol (**2**), carnosol (**3**), carnosic acid (**4**), and the triterpene betulinic acid (**5**) were able to block the activation of AhR by TCDD. However, the current study did not investigate the effects of the isolated AhR antagonists on the AhR expression. Previous studies have shown that BA can increase AhR expression through demethylation on the AHR promoter [62].

Apigenin acts as an antagonist of AhR [63,64], while DMA acts as a partial agonist. It, therefore, appears that double methylation of apigenin in DMA leads to a change in its behavior over AhR: from antagonist to partial agonist (DMA). In addition, the presence of the methoxy group (-OCH3) at position seven of the carnosol molecule to form 7-*O*-methyl-epi-rosmanol resulted in the conversion of the action of the starting compound from antagonist against to partial agonist regarding AhR.

### 3.2. AhR and Skin Cancer

It is well established that skin cancer is the most common type of cancer in Caucasians [65], and AhR seems to play an important role in carcinogenesis as well as in the development of various types of skin cancer. AhR was identified as a new risk factor for cutaneous squamous cell carcinoma (SCC) in a genome-wide association study [66]. UVR activated AhR acts as a light sensor in keratinocytes. Furthermore, UVR (especially UVB) generates FICZ, a tryptophan derivative, in these cells [6], which in turn binds to AhR, activating it. This is believed to be one of the mechanisms that UVR increases the risk of skin cancer via attenuation of the DNA repair system and apoptosis along with enhancement of the UV response [22], allowing UV damaged cells to survive and become carcinogenic. In addition, it has been shown that AhR deficient mice are more protected against UV-induced skin cancer [20].

As also mentioned in previous studies, AhR activation by air pollutants is found to be related to skin carcinogenesis [17]. It has been demonstrated that i.p. or s.c. administration of a total of 600 μg/kg of body weight TCDD resulted in facial region skin cancer development in 21% of the hamsters participating in the experiment. The neoplasms developed within 12–13 months from the initial administration while no other organ was affected [13]. Further experiments have demonstrated the role of AhR activation in PAH-induced skin carcinogenesis. Chronic topical application of organic extracts of airborne particulate matter (PM) results in skin cancer development in half of the AhR+/+ mice but in none of the AhR−/− ones [17]. Topical or subcutaneous application of benzo[a]pyrene (another PAH contained in PM from cigarettes or air pollutants) to wild-type mice may also cause SCC, while this carcinogenic result is attenuated when applied to AhR deficient mice [14].

In total, environmental factors that induce skin carcinogenesis via AhR activation include: (a) ultraviolet radiation (UVR), which is associated with the majority of melanoma cases (65%) as well as with almost all other skin cancer cases (90%) including BCC and SCC [67], given the fact that partial mediation of ultraviolet radiation damage became through the photochemical intracellular production of the potent AhR ligand (FICZ) [6,18,21], (b) carcinogenic chemicals in air pollutants as TCDD, and (c) metabolites of *Malassezia* yeast implicate in extramammary Paget’s disease (EMPD) [68]. The association between *Malassezia*-produced indoles and BCC is considered to be the result of AhR mediated local metabolic and immune aberrations [18], while ICZ (like TCDD) are considered tumor inducer substances [15]. Moreover, UVB-induced skin damage prevention through chemical inhibition of AhR signaling in human skin could minimize the development (appearance) of non-melanoma skin cancer [69].

AhR activation by polycyclic aromatic hydrocarbons (PAHs) leads to increased expression of CYP1A1 and CYP1B1, which transform PAHs into genotoxic metabolites. Carnosol (CS) can inhibit Hsp90 ATPase resulting to lower levels of AhR and causing a reduction of B[a]P-mediated induction of CYP1A1 and CYP1B1 and finally blocking the formation of DNA adducts in HaCaT cells [34].

RO and its constituent carnosol have been found to inhibit tumor initiation and promotion. The topical application of carnosol has been found to inhibit tumor development. RO application to mouse skin prevented the binding of benzo[a]pyrene to skin cell DNA, thus inhibiting tumor initiation by B(a)P and DMBA [32]. Moreover, a RO 65% (*v*/*v*) hydroalcoholic extract was found to be able to reduce the proliferation of human melanoma A375 cells (which are resistant to cytotoxic agents) [33].

In a DMBA induced skin cancer nude mouse model, oral administration of 500 or 1000 mg/kg/day of ROE for 15 weeks resulted in a significant decrease in tumor development, including number, diameter, and weight, while an increase in latency period compared to control population was observed [70,71].

Additionally, in several other skin cancer models (including Ht-1080, BEAC, HUVEC, and B16F10 cells), carnosic acid was found to inhibit cell survival, cell migration, and cell adhesion and enhanced apoptosis and induced cell-cycle arrest [72,73]. Topical application of known AhR antagonists, as some plant polyphenols (resveratrol, epigallocatechin-3-gallate), was found to have a protective role against UVB-induced skin cancer in mice [74,75,76]. As referred previously, ROE could protect from skin carcinogenesis caused by environmental factors. Now, with our study, it seems that ROE potentially could prevent skin cancer caused by the exposure (1) to UV radiation, (2) to metabolites of *Malassezia* yeast, or (3) to environmental pollutant TCDD via the antagonistic effect on AhR with respect to FICZ, IND, and TCDD. Therefore, we add a new potential way for the chemopreventive and anticancer action of ROE. In addition, carnosic acid, carnosol, and betulinic acid all showed a strong dose-depended antagonist activity versus TCDD on AhR.

Corre et al. [77], pustulated that sustained activation of AhR plays a dominant role in the development of resistance to BRAF inhibitors (BRAFi) by melanoma cells. Resveratrol, a clinically compatible AhR antagonist that eliminates its prolonged harmful activation, in combination with BRAFi, has been shown to reduce the number of BRAFi-resistant cells and delay tumor growth. Therefore, the authors proposed the weakening of AhR action as a strategy to address BRAFi resistance and recurrence in melanoma.

Furthermore, activation of AhR appears to favor the maintenance of melanoma by increasing the density of programmed cell death protein 1 (PD-1) in tumor-infiltrating T cells surrounding melanoma, resulting in cancer cells escaping the immune response [78].

Hence, the development of therapeutic agents that regulate AhR activity is a promising strategy for the prevention and treatment of skin cancers. The leaf extract of *R. officinalis* L. and its substances CA, CS, and BA as antagonists of AhR inhibit its prolonged activation and could be useful candidates for the adjuvant treatment of melanoma (1) together with BRAFi and (2) together with immunotherapy drugs, such as monoclonal antibodies that bind PD-1 or PD-L1.

### 3.3. Perspectives

It can be highlighted that according to EMA (European Medicine Agency, HMPC Herbal Medicinal Products Committee) and the FDA (Food and Drug Administration) of the USA, rosemary has extremely low toxicity and has been classified as GRAS (Generally Recognized as Safe) and approved as an “antioxidant food preservative” [79,80].

Concerning previous studies regarding ROE and AhR, Amakura et al. [81] found that the ethyl acetate extract of rosemary elicited notable AhR activation, and they identified some AhR-activating substances in this sample. Among them, nepitrin and homoplantagenin, which are flavone glucosides, showed a dose-related AhR-binding activity. This study is in complete disagreement with our results. However, none of these substances were the same as ours. It seems that their rosemary extract was prepared in a completely different way. For example, our extracts did not contain rosmarinic acid, caffeic acid, and glucosides, and their extract did not contain abietinic diterpenes or triterpenes. However, this is a good example of the chemical diversity and the bimodal action of many plants extracts: same plant, same drug, different manipulation, different extract, different action.

ROE, as it usually happens with plants extracts, does not behave as a single drug, but as a multifactorial potential therapeutic agent, due to the synergistic and various effects of its ingredients. A multifactorial disease such as SD could be treated better with a multifactorial remedy than with a drug having high specificity about one single molecular target. A natural product can play an important role, acting in a pleiotropic way, offering a good multifactorial remedy.

## 4. Materials and Methods

### 4.1. Chemicals

TCDD was obtained from Dr. Stephen Safe (Texas A&M University), [^3^H]TCDD (13 Ci/mmol) was obtained from ChemSyn Laboratories (Lenexa, KS, USA), and 2,3,7,8-tetrachlorodibenzofuran (TCDF) was from Accustandard (New Haven, CT, USA). [^32^P]-ATP (~6000 Ci/mmol) was from Perkin Elmer Life and Analytical Sciences. Dimethyl sulfoxide (DMSO) was from Sigma-Aldrich (St. Louis, MO, USA). FICZ, IND, PZ were synthesized or isolated as previously described [11,30].

### 4.2. Plant Material

*R. officinalis* L. leaves were collected from the botanical garden of the National and Kapodistrian University of Athens. The leaves were dried at room temperature for 14 days.

### 4.3. Extracts Preparation

Five different dry extracts from rosemary leaves (ROS) were prepared using different solvents or times of extraction (Table 1). Briefly, the preparation method was the following: 100 g of the powdered dry leaf of rosemary was treated with each solvent, in a 1:10 ratio (100 g of dry leaf and 1 L of solvent), for a specific period in an opaque recipient. At the appropriate time, the plant material was separated from the liquid part by filtration. Then, the volume of the solution was restored to the original volume (1 L), and 2 L of distilled water was added. The formed precipitate was collected by filtration, dried, and finally powdered to afford the corresponding rosemary dry extract.

### 4.4. Isolation of ROE Ingredients

By using normal phase column chromatography and preparative thin-layer chromatography for the final purification, the following compounds were isolated from RO leaves: 4′,7-*O*-dimethylapigenin (**1**), 7-*O*-methyl-*epi*-rosmanol (**2**), carnosol (**3**), carnosic acid (**4**) and betulinic acid (**5**). The identity of each compound was confirmed by 1D and 2D NMR spectroscopy and comparison with bibliographic data [82,83,84,85]. Details of the isolation procedure and NMR data are given in the Appendix A.

For quantitative analysis, a calibration curve for isolated abietane diterpenes and for BA and DMA was constructed based on selected non-overlapping ^1^H-NMR signals: carnosic acid (y = 0.4812x − 0.0067, R^2^ = 0.9999), carnosol (y = 0.4565x − 0.0043, R^2^ = 0.9999), 7MER (y = 0.4917x − 0.0025, R^2^ = 0.9999), DMA (y = 0.2087x − 0.0109, R^2^ = 0.9998), BA (y = 0.624x + 0.0012, R^2^ = 0.9998). Examples of NMR spectra of the extracts showing the peaks used for quantitation as well as the quantitation results for each extract are given in the Appendix A.

### 4.5. NMR Spectroscopy

All experiments were performed on a Bruker Avance DRX 500 MHz NMR spectrometer (Bruker Biospin, Rheinstetten, Germany) operating at NMR frequency of 500.13 MHz for ^1^H and 125.77 MHz for ^13^C NMR. All ^1^H (500 MHz) and ^13^C NMR (125 MHz) spectra were recorded with chemical shifts in δ (ppm) and coupling constants (J) in hertz (Hz).

### 4.6. Preparation of Cytosol

Hepatic cytosol was prepared in HEDG buffer (25 mM Hepes (pH 7.5), 1 mM ethylenediaminetetraacetic acid, 1 mM dithiotreitol, and 10% (*v*/*v*) glycerol). Protein concentrations were determined by dye binding using bovine serum albumin as the standard, and cytosol was stored frozen at −80 °C until use, as previously described [86].

### 4.7. Hydroxyapatite AhR Ligand Binding Assay

Aliquots of guinea pig hepatic cytosol (2 mg/mL) were incubated with 2 nM [^3^H]TCDD in the presence of DMSO (1%), TCDF (200 nM), or the indicated chemical or sample extract in DMSO (1%) for 2 h at 20 °C in a water bath. [^3^H]TCDD binding in aliquots of the incubation (200 μL) was determined by HAP binding as previously described in detail [87]. The total amount of [^3^H]TCDD specific binding was obtained by subtracting the nonspecific binding ([^3^H]TCDD and TCDF) from the total binding ([^3^H]TCDD). The ability of a chemical(s) in a sample extract to bind to the AhR was indicated by its ability to competitively reduce [^3^H]TCDD specific binding. The amount of [^3^H]TCDD specific binding remaining in the presence of competitor chemical was expressed as a percent of the total [^3^H]TCDD specific binding.

### 4.8. Gel Retardation Assay

Complementary synthetic oligonucleotides containing the DRE3 AhR DNA binding site (i.e., dioxin responsive element 3 from the upstream region of the murine CYP1A1 gene (XX)) (5′-GATCTGGCTCTTCTCACGCAACTCCG-3′ 5′-CAACTCCGGATCCGGAGTTGCGTGAGAAGAGCCA-3′ were prepared, reannealed, and end-labeled with [^32^P]ATP as described [87]. Guinea pig hepatic cytosol (8 mg/mL in HEDG) was incubated for 2 h in a room temperature water bath with DMSO (2% final concentration), TCDD (20 nM final concentration in DMSO), or 20 nM TCDD in the presence of the indicated concentration of extract or chemical. After incubation, an aliquot of the reaction was mixed with poly[dI**•**dC] and [^32^P]-DRE (100,000 cpm), and AhR:DRE complexes were resolved by gel retardation analysis as described in detail [88], visualized using a FLA9000 Fujifilm Imager (Walnut Creek, CA, USA) and protein-DNA complexes quantitated with Fujifilm MultiGauge software.

### 4.9. AhR-Responsive Luciferase Reporter Gene Analysis

Recombinant human hepatoma (HG2L7.5c1) cells containing a stably integrated AhR-responsive luciferase reporter gene plasmid (pGudLuc7.5) with 20 dioxin-responsive elements (DREs) were plated (75,000 cells/well) into white, clear-bottomed 96 well tissue culture plates in 100 μL α-MEM containing 10% FBS and allowed to attach for 24 h at 37 °C. Cells were incubated with carrier solvent DMSO (1% final concentration), 10 nM TCDD (in DMSO), 100 ppm rosemary extract, or TCDD, and 100 ppm rosemary extract (for antagonism analysis) for 24 h at 37 °C. After the incubation, cells were visually inspected for toxicity, washed with phosphate-buffered saline, followed by addition of 50 μL of passive lysis buffer (Promega), and cells were lysed 20 min at room temperature with shaking. Luciferase activity (expressed as relative light units (RLUs)) in each well was measured by integrating luminescence over 10 s with a 10 s delay in an Orion microplate luminometer (Berthold Detection Systems, Bad Wildbad, Germany) following automatic injection of Promega stabilized luciferase reagent.

### 4.10. Quantitative Real-Time Polymerase Chain Reaction

Human keratinocytes (SIK 28) cells grown to confluence as previously described [89] in 6-well plates were incubated with DMSO (1% (*v*/*v*)), or 20 nM of TCDD, PZ, FICZ, or IND (in DMSO 1% (*v*/*v*)) in the absence or presence of the methanolic extract R4 (10 ppm) for 2 h. After incubation, cells were washed with PBS, total RNA isolated using TRIzol reagent (ThermoFisher Scientific, Waltham, MA, USA), and cDNA synthesized using the High-Capacity cDNA Reverse Transcription Kit (Applied Biosystems, San Francisco, CA, USA) with a Bio-Rad T100 Thermal Cycler. cDNAs were quantitated using Applied Biosystems Taqman gene expression assays for human CYP1A1 (ID: Hs00153120_m1), and the housekeeping gene β-glucuronidase (GUSB; ID: Hs99999908_m1) with an Applied Biosystems 7500 Fast Sequence Detection System (ThermoFisher Scientific) as previously described [38,90]. CYP1A1 mRNA levels were normalized to those of GUSB, and values expressed relative to GUSB levels in DMSO-treated cells (set to a value of 1.0) as per the delta-delta Ct (2^−^^ΔΔCT^) method [91].

### 4.11. Statistical Analysis

All independent experiments were performed in triplicate, and the results are expressed as mean ± SD. Descriptive statistics, analysis of variance (ANOVA), and Duncan’s multiple range test (MRT) at *p* ≤ 0.01 were performed in order to check the hypothesis that there were statistically significant differences among the treatments. Data in percentages were subjected to appropriate log or arcsine transformation for proportions before statistical analysis and were transformed back to percentages for presentation in Graphs. All statistical analysis was performed using SPSS v.20 software for Windows (IBM SPSS Statistics 2011, IBM Corp., Armonk, NY, USA).

## 5. Conclusions

The results of this study demonstrate for the first time that *Rosmarinus officinalis* L. leaf extract shows high antagonist activity in vitro (guinea pig cytosol) and in human keratinocyte cells (CYP1A1 mRNA induction) against various agonists including TCDD, IND, FICZ, and PZ. Moreover, metabolites of *R. officinalis* L., as abietane derivatives carnosol, carnosic acid, and triterpene betulinic acid, exhibited dose-dependent antagonist activity against TCDD in vitro (guinea pig cytosol), while diterpene 7-*O*-methyl-epi-rosmanol and flavone 4′,7-*O*-dimethylapigenin in the same experiment behaved as partial agonists. ROE-mediated inhibition of AhR activity is likely to contribute to the chemopreventive, anticancer, and anti-inflammatory activity of the herb. Hence, regarding skin conditions, ROE could prevent dermal cell damage induced by UV radiation via FICZ, environmental pollutants such as TCDD, and the *Malassezia* metabolites, such as FICZ, PZ, and IND. These observations support the potential use of ROE and/or herbal crude material for the protection of skin health. Further in vivo and clinical studies should be performed in this direction.

## Figures and Tables

**Figure 1 molecules-27-02499-f001:**
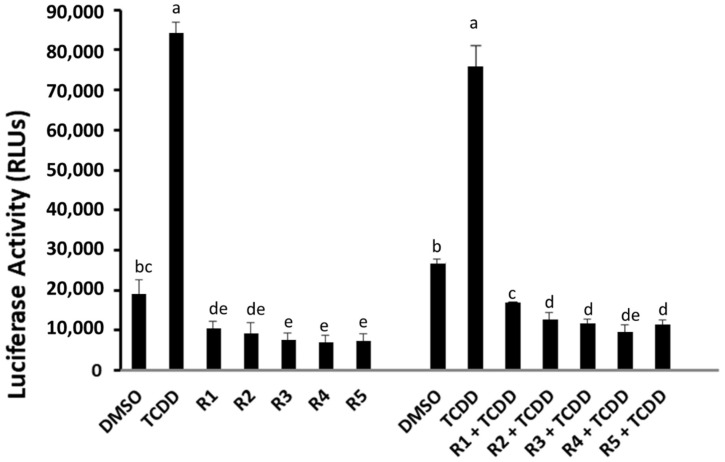
In vitro luciferase activity in human hepatoma cells (HG2L7.5c1) incubated with TCDD (1 nM) and with each of the R1–R5 rosemary extracts alone (100 ppm each) to investigate the presence of agonist activity on AhR, and with the combination of TCDD with each of the R1–R5 rosemary extracts (R 100 ppm + TCDD 1 nM) to investigate inhibitory activity. The bar means followed by the same letter (as superscript) are not statistically different at *p* ≤ 0.01.

**Figure 2 molecules-27-02499-f002:**
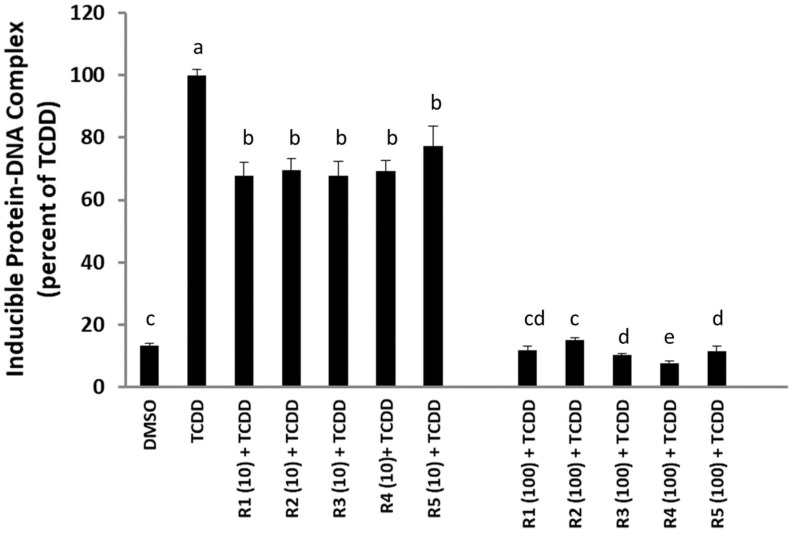
Gel retardation assay (GRA) in guinea pig hepatic cytosol to evaluate the percentage of AhR-DNA complex formed when each rosemary extract (R1–R5, at two different concentrations of 10 and 100 ppm) is incubated together with TCDD (1 nM) for 2 h. The value is set to 100% when there is only TCDD. The bar means followed by the same letter (as superscript) are not statistically different at *p* ≤ 0.01.

**Figure 3 molecules-27-02499-f003:**
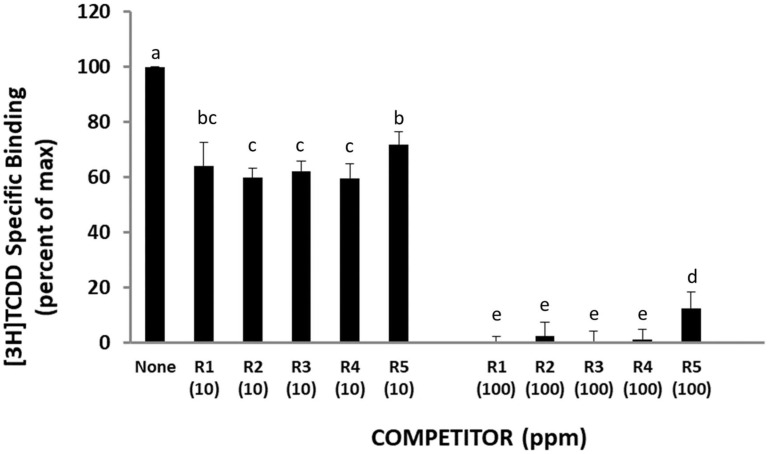
Rosemary extracts activities as antagonists of AhR ligand binding in guinea pig hepatic cytosol in the presence of 2 nM [^3^H] TCDD for 2 h. For R1–R5 abbreviations see Table 1. The bar means followed by the same letter (as superscript) are not statistically different at *p* ≤ 0.01.

**Figure 4 molecules-27-02499-f004:**
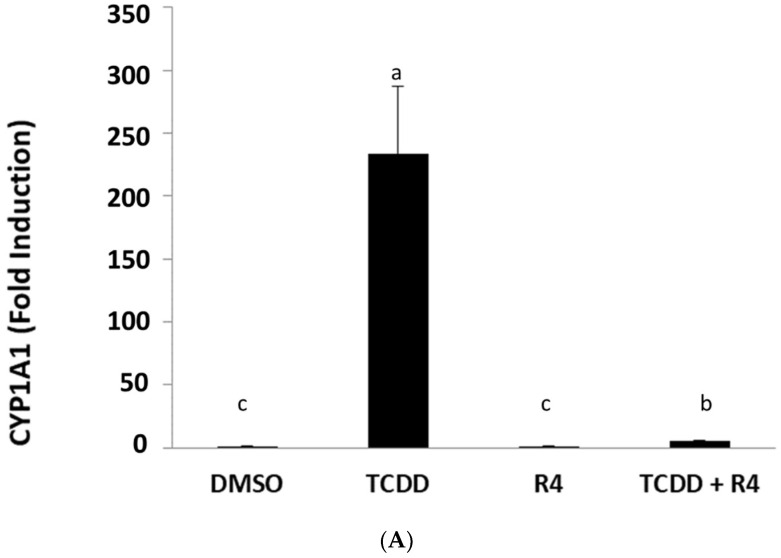
(**A**,**B**) Action of the methanolic extract-R4, at 10 ppm, in human keratinocytes (SIK 28) after 2 h of incubation, in absence and in presence of TCDD or PZ or FICZ or IND. The vertical axis represents the CYP1A1 mRNA induction measured by using RT-PCR. The solvent was dimethylsulfoxide (DMSO). Results are expressed as means ± SD. The bar means followed by the same letter (as superscript) are not statistically different at *p* ≤ 0.01.

**Figure 5 molecules-27-02499-f005:**
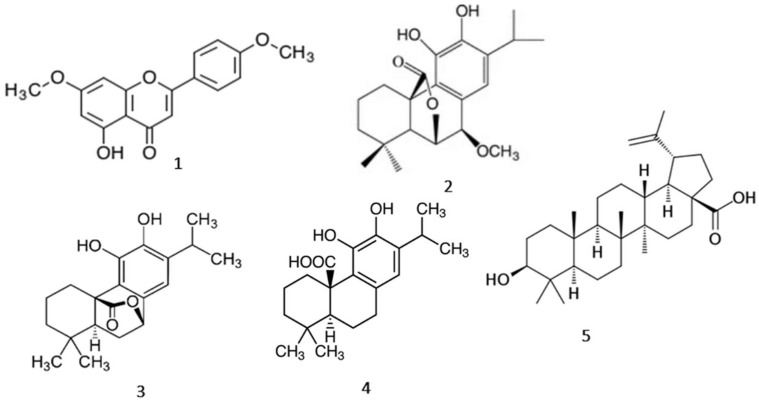
The structures of major metabolites isolated from extract RO-4: 4′,7-*O*-dimethylapigenin (DMA) (**1**), 7-*O*-methyl-*epi*-rosmanol (7MER) (**2**), carnosol (CS) (**3**), carnosic acid (CA) (**4**), betulinic acid (BA) (**5**).

**Figure 6 molecules-27-02499-f006:**
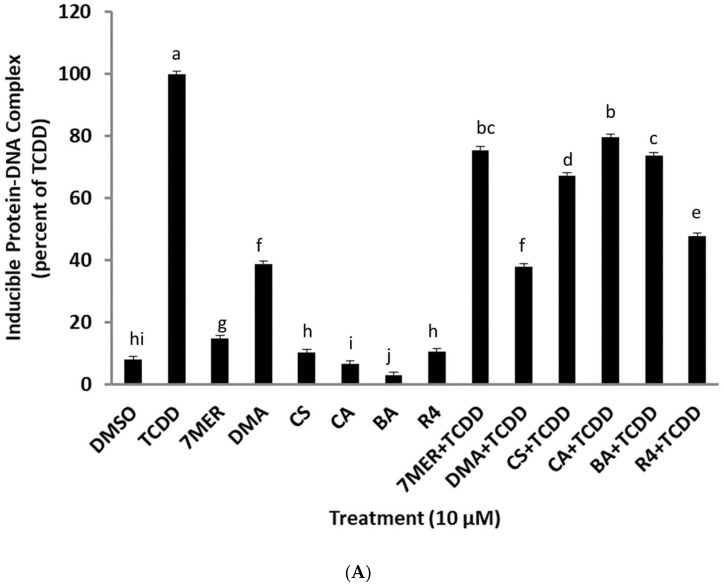
(**A**,**B**) Gel retardation assay (GRA) in guinea pig cytosol for the investigation of the behavior of CS, CA, 7MER, DMA, BA and R4 versus AhR in the presence and absence of 20 nM of TCDD. Time of incubation 1.5 h. Results are expressed as means ± SD. (7MER = 7-*O*-methyl-*epi*-rosmanol, DMA = 4′,7-*O*-dimethylapigenin, CS = carnosol, CA = carnosic acid, BA = betulinic acid, R4 = total methanolic extract). The bar means followed by the same letter (as superscript) are not statistically different at *p* ≤ 0.01.

**Table 1 molecules-27-02499-t001:** Solvents, ratio (Drug/Solvent) and time of maceration for extracts of *Rosmarinus officinalis* L. dry leaves.

Rosemary Extract	Solvent	Drug/Solvent Ratio (*w*/*w*)	Time of Maceration	Yield (%, *w*/*w*)
R1	Ethanol 96°	1:10	48 h	8.64
R2	Ethanol 96°	1:10	14 days	7.14
R3	Methanol	1:10	48 h	10.13
R4	Methanol	1:10	7 days	9.86
R5	Isopropanol	1:10	14 days	5.41

## Data Availability

The data presented in this study are available on request from the corresponding author.

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
