# Peer review of "Rosmarinus officinalis L. Leaf Extracts and Their Metabolites Inhibit the Aryl Hydrocarbon Receptor (AhR) Activation In Vitro and in Human Keratinocytes: Potential Impact on Inflammatory Skin Diseases and Skin Cancer"

_molecules, 2022, doi:10.3390/molecules27082499_

Round 1

Reviewer 1 Report

Journal: Molecules

Manuscript ID: molecules-1638006

Title: Rosmarinus officinalis L. leaf extracts and their metabolites, inhibit the aryl hydrocarbon receptor (AhR) activation in vitro and in human keratinocytes: potential impact on inflammatory skin diseases and skin cancer

Authors: Panagiotis Kallimanis, Ioanna Chinou, Angeliki Panagiotopoulou, Anatoly A. Soshilov, Guochun He, Michael S. Denison, Prokopios Magiatis*

  1. The manuscript focused on the identification of natural compounds that could inhibit the AhR activation by ligands of microbial origin. Also, the methanolic extract of Rosmarinus officinalis L. leaf is used for CYP1A1 mRNA induction study using RT-PCR, in human keratinocytes against TCDD, FICZ, PZ, and IND.
  2. The research gap and goals of the study are well explained in the introduction section
  3. The present study will be helpful for the treatment of skin diseases and skin cancer by plant extracts.
  4. Authors conclude that the Rosmarinus officinalis L. could be useful for the prevention or treatment of skin diseases mediated by activation of AhR.

Recommendations:

1) If possible add the images of In-vitro luciferase activity in human hepatoma cells (HG2L7.5c1) in figure 1.

The authors have presented the article in a very nice way; the work done is to the mark of the journal. Therefore, I recommend the acceptance of the article.

Author Response

We thank reviewer 1 for its positive comments.

Concerning the proposed recommendation there are no available images to be added to figure 1. The figure represents the measured luciferase activity in human hepatoma cells (HG2L7.5c1). Luciferase activity (expressed as relative light units (RLUs)) in each well was measured (integrating luminescence over 10 s with a 10 s delay) in an Orion microplate luminometer following automatic injection of Promega stabilized luciferase reagent.

Reviewer 2 Report

Comments to the Author

The content of this manuscript is well-designed and well-explains the purpose, method, and results of the study. In addition, results strongly supports Rosmarinus officinalis L. leaf extracts and their metabolites, inhibit the AhR activation via experimental verification.

There are some minor points which need to be considered for improving clarity of the MS.

1. Please describe the statistical processing method in the ‘’Materals and Methods’ section.

2. A standard deviation error bar display is required in Fig.1.

3. All graph figures need explanation for statistical significance.

4. In Table 1, it would be better if the yield was also indicated.

Author Response

  1. The following paragraph “4.11 Statistical analysis” has now been added in Materials and methods. All independent experiments were performed in triplicate and the results are expressed as mean±SD. Descriptive statistics, analysis of variance (ANOVA) and Duncan’s multiple range test (MRT) at p ≤ 0.01 were performed in order to check the hypothesis that there were statistically significant differences among the treatments. Data in percentages were subjected to appropriate log or arcsine transformation for proportions before statistical analysis and were transformed back to percentages for presentation in Graphs. All statistical analysis was performed using SPSS v.20 software for Windows (IBM SPSS Statistics 2011, IBM Corp., Armonk, NY, USA).
  2. The standard deviation error bars have been added in Figure 1.
  3. Explanations for statistically significant differences have been added in all figure legends.
  4. Yields have been added in Table 1.

Reviewer 3 Report

The manuscript titled “Rosmarinus officinalis L. leaf extracts and their metabolites, inhibit the aryl hydrocarbon receptor (AhR) activation in vitro and in human keratinocytes: potential impact on inflammatory skin diseases and skin cancer” presents the inhibition of aryl hydrocarbon receptor by various Rosmarinus officinalis extracts (ROE), including ethanol (R1,R2), methanol (R3,R4) and isopropanol (R5) extracts. Moreover, several compounds were isolated and characterized in these extracts, including 4’,7-O-dimethylapigenin (1), 7-O-methyl-epi-rosmanol (2), carnosol (3), carnosic acid (4) and betulinic acid (5). The study comprehends various complementary analyses that reveal the antagonistic role of ROE and some of its components in relation to AhR activation eg. by the dioxin pollutant TCDD. It is known from the literature that AhR activation can contribute to skin carcinogenesis, via different mechanisms, eg. inhibition of apoptosis, so the present manuscript contains relevant information in this context. The study in its present form contains, nevertheless, important limitations that require major revision. The major points of discussion are:

  1. number of independent experiments: the authors do not mention the number of independent experiments performed for each assay. This information needs to be provided at least in each figure that relates to experimental assays, as well as results from statistical comparison between groups or samples vs. control;
  2. DMSO effect: in mouse embryonic fibroblasts, DMSO is reported to activate the AhR receptor, eg. Wuputra et al., 2021 (doi: 10.1007/s10565-021-09592-2). In the present manuscript, it is not clear what kind of controls were run to compare DMSO-free system with 1% and 2% DMSO, particularly considering these experiments were performed in 1% and 2% DMSO.
  3. Effect on AhR expression: betulinic acid can increase AhR receptor expression through demethylation on the AHR promoter, eg. doi: 10.18632/oncotarget.21889; the described effects of AhR antagonists should be framed with their effects on AhR expression, as described in the literature (increased or decreased expression)

Minor points include:

Lines 470-471: Preparation of Cytosol – please indicate the method used for determination of protein concentrations.

Concentration units of AhR antagonists are not consistent throughout the different assays: in some assays, ppm is used, while in other assays micrograms/ml. Please consider uniformization.

Author Response

  1. All independent experiments were performed in triplicate. This information has been added in the new paragraph statistical analysis 4.11
  2. Although it is known that DMSO can induce a low activation of AhR we did not perform control tests for DMSO-free samples. In all cases, the tested compounds were dissolved in the same final concentration of DMSO, so the actual control was the sample that was tested solely with DMSO but without the addition of any of the tested compounds or extracts. Actually, it was not possible to add the tested compounds or extracts without the use of DMSO, so for this reason the DMSO-free test had not any actual usefulness. This is now more clearly described in paragraphs 4.7 and 4.10
  3. A relative comment has been added in line 362-365 and a new reference [62]

Minor points

The method used for determination of protein concentrations was the Bradford method as described in reference 86

The concentration units have been now written in uniform way as ppm and not as microgram/ml

Reviewer 4 Report

The authors state the following: "ROE could be useful for the 30
prevention or treatment of skin diseases mediated by activation of AhR". I think this is the great limitation of this paper: The study is based on an extract and therefore, the biological activity cannot be associated with any specific compound.

In Medicinal Chemistry drugs are studied, which are the structures responsible for biological activity and never mixtures of them. Yes, it is true that the borders between the different disciplines are less and less clear, but I advise authors to send this article to a journal focused on pharmacognosy or even pharmacology.

Author Response

We are sorry but most probably there is a big misunderstanding. As we clearly explain in the manuscript, the initial inhibitory activity was recorded for the Rosemary extracts but then we continued to the isolation and quantitation of pure ingredients of rosemary extracts that then were evaluated for their activity, especially carnosol, carnosic acid and betulinic acid. For all the isolated compounds we performed a study of their inhibitory activity on AhR against a series of well-known ligands proving that the activity of the extract is related at least with some of the studied isolated compounds.

Round 2

Reviewer 3 Report

The revised version of the manuscript is more clear concerning experimental replication and statistical analysis. Nevertheless, DMSO solvent concentrations used were too high for use in biological systems (well above 0.1%). It is not trivial that using 1% or 2% DMSO solvent as control obviates the issue of intereference with the action of potential AhR inhibitors. If there is a large difference between 0% and 1% or 2% DMSO in AhR activity this will limit the validity of results, thus resultus of AhR activity in 0% versus 1 and 2% DMSO needs to be included.

Additionally, the SIK28 cell line indicated is not mentioned in the referenced work (ref. 89). The origin of this cell line, as well as culture conditions (cell culture medium, temperature etc) must be disclosed.

Reviewer 4 Report

After the clarifications and modifications carried out by the authors, the article can be published in Molecules.